# OpenReview forum: "Can Video LLMs Refuse to Answer? Alignment for Answerability in Video Large Language Models"
_ICLR.cc/2025/Conference — ICLR 2025 Poster_

### Official Review · Reviewer_jkxc · 2024-10-28

**Soundness:** 3
**Presentation:** 4
**Contribution:** 3
**Rating:** 6
**Confidence:** 4

**Summary:**

This paper addresses the issue of "answerability" in Video Large Language Models (Video-LLMs). Current Video-LLMs excel at answering questions directly related to the content of a video, but struggle with questions that go beyond the video's scope, often hallucinating plausible but incorrect answers. The authors propose a framework called "alignment for answerability" to train Video-LLMs to recognize and refuse to answer such unanswerable questions.

This framework involves:
- Alignment:  Training Video-LLMs to assess the relevance of a question to the video content and respond with "unanswerable" when appropriate.  They formally define this alignment process and associated scoring functions.
- Evaluation Metrics:  Beyond simple accuracy, the authors propose a set of metrics to comprehensively evaluate the alignment process. These include measuring how often the model refuses to answer valid questions (Excessive Refusal), how often it correctly answers previously refused questions (Permissiveness), and how well it identifies and declines truly unanswerable questions (Discretion).
- Dataset Creation (UVQA): The authors created a new dataset, UVQA, specifically for training and evaluating answerability.  They leverage existing video-description datasets, altering the descriptions to generate questions that are unanswerable based on the accompanying video.  They further categorize these questions based on the type of mismatch (object, attribute, or relationship).

Experiments demonstrate that models trained with this framework and the UVQA dataset significantly improve their ability to handle unanswerable questions. The authors also explore an alternative approach based on decomposing questions into multiple existence-based sub-questions (inspired by the POPE approach for image-based LLMs), but find it less effective and computationally more expensive than their proposed framework.

**Strengths:**

- The paper tackles an important yet often overlooked issue of answerability in Video-LLMs.
- The paper defines sound metrics and data generation pipeline.
- The results on the in-domain evaluation set are convincing, notably with a positive comparison against the POPE baseline.

**Weaknesses:**

- The evaluation set is curated from the same distribution as the training set, which may overestimate the benefits and results of the alignment procedure. It would be great to see the results on a fully human-annotated set, ideally on videos from a different distribution (e.g. using videos from MSR-VTT).
- Having results on academic VideoQA benchmarks (e.g. the full Activitynet-QA evaluation set) would enable to double check the absence of accuracy degradation compared to the state of the art. Has any trade-offs between answerability performance and standard VideoQA performance been observed?

**Questions:**

- The approach could be extended to images in future work. Is there any challenge the authors anticipate in doing so?

---

> ### Author Response · Authors · 2024-11-24
> **Author Response**
>
> ### [Response to Weaknesses & Questions]
> ### **Weakness 1: Evaluation on fully human-annotated set with videos from a different distribution.**
>
> Thank you for this suggestion. **We want to note that the current UVQA dataset has no video overlap between the training and evaluation sets.** That being said, we agree that the evaluation of different video sources will be a valuable addition to our experiments. Thus, we are currently in the process of annotating unanswerable question-and-answer pairs on videos from MSR-VTT. **We will update the results once they are available.**
>
> ---
>
> ### **Weakness 2: Trade-offs between answerability performance and standard VideoQA performance.**
>
> As is often the case in multi-objective optimization, there is a trade-off in our setup as well. This trade-off is an inherent limitation, which we have explicitly acknowledged and addressed in our work. The metrics we propose, such as  $S\_{\text{ex-ref}}$ (Eq. 5), are designed to evaluate models while considering such trade-offs. For example,  $S\_{\text{ex-ref}}$  quantifies the extent to which models, after alignment, decline to answer answerable questions that it was previously capable of answering correctly. A lower $ S_{\text{ex-ref}}$  score is better, and unaligned models naturally achieve a score of 0. However, after alignment through SFT or DPO, models exhibit  $S\_{\text{ex-ref}} > 0$, indicating some trade-off in performance on previously answerable questions.
>
> To further address this, we added a section in **[Appendix A.10: Pareto Front Analysis]** to comprehensively illustrate the trade-off. Specifically, we observe that alignment for answerability improves performance on the unanswerable UVQA dataset but slightly decreases accuracy on the answerable QA dataset. Notably, the Pareto front analysis demonstrates that the aligned models (SFT, DPO) achieve a better balance, positioning them at an optimal point where performance on unanswerable questions improves without excessively compromising performance on answerable questions.
>
> ---
>
> ### **Question 1: Extension to images**
>
> Our framework can easily be applied to image-based vision-language models (VLMs) since images can be treated as a single frame of a video. However, we would like to note that while some prior research [1, 2] has addressed unanswerability in image-based models, to the best of our knowledge, there has been little to no investigation into this issue within open-ended QA tasks for Video-LLMs. This gap highlights the importance of our contribution. According to [1, 2], image-based VLMs demonstrate approximately 12% accuracy in unanswerability detection under zero-shot settings. In contrast, as shown in Figure 1-(a) and (b) of our paper, open-source Video-LLMs exhibit near-zero capability in detecting unanswerability. This indicates that the challenge is significantly more pronounced in the video domain, making it a critical issue to address.
>
> ```
> [1] Whitehead S, Petryk S, Shakib V, et al. Reliable visual question answering: Abstain rather than answer incorrectly, ECCV 2022
> [2] Guo Y, Jiao F, Shen Z, et al. UNK-VQA: A Dataset and a Probe into the Abstention Ability of Multi-modal Large Models. TPAMI
> ```

---

> > ### Comment · Reviewer_jkxc · 2024-11-25
> >
> > I thank the authors for their detailed answer.
> >
> > Weakness 1: Thank you for running this experiment, I am looking forward to seeing the results (as well as the results to the related Weakness 5 from Reviewer GEbz).
> >
> > As for Weakness 2 and Question 1, I appreciate the answers and hope they will be included in the final version.

---

> ### Author Response · Authors · 2024-11-27
> **Author Response (Follow Up on Human-annotated Set Evaluation)**
>
> Thank you for your patience. We have finished the evaluation on fully human annotated set from videos from a different source. The relevent section has been added to ***[Appendix A.12: Out-of-Distribution Evaluation Set]***. We used the MSR-VTT [1] dataset for the video source and created human-labeled unanswerable QA pairs for the evaluation.
>
> In ***Table 6 of Appendix A.12***, we show that the aligned model outperforms the unaligned model even on the out-of-distribution evaluation set, highlighting its ability to generalize beyond the training distribution. Although the performance gains are somewhat smaller compared to the in-distribution evaluation results in Table 1, the aligned model consistently demonstrates improved performance.
>
> > [1] Xu et al. "MSR-VTT: A Large Video Description Dataset for Bridging Video and Language", CVPR 2016
>
> Moreover, we have also addressed the *Weakness 5 from Reviewer GEbz (Human Evaluation)* and added this into ***[Appendix A.11: Human Evaluation of Model Prediction]***. The result from the human evaluation confirms that the aligned model, using our framework, outperforms the unaligned model.
>
> We sincerely thank the reviewer once again for the time and thoughtful feedback.

---

> > ### Comment · Reviewer_jkxc · 2024-11-27
> >
> > I appreciate these additional experiments and confirm that I support this paper for acceptance.

---

### Official Review · Reviewer_yiLe · 2024-10-29

**Soundness:** 3
**Presentation:** 3
**Contribution:** 3
**Rating:** 6
**Confidence:** 4

**Summary:**

The paper aims to improve Video-LLMs’ capabilities of differentiating visually-unaligned questions, while maximally retain the performance on answering common answerable questions.
Specifically, the paper first defines a suite of metrics to evaluate alignment for answerability. It then collects data for both training and evaluation. The experiments show that existing Video-LLMs almost cannot refuse to answer unaligned questions. While SFT and DPO can effectively improve the overall performance, DPO significantly outperforms SFT in maintaining the models’ original performance while handling unaligned questions.

**Strengths:**

1.	The work is overall well-developed and shows the authors’ insights in the problem.
2.	The constructed benchmark could be valuable, with dedicated evaluation metrics and annotations (reason for unanswerable).
3.	The paper implements both SFT and DPO for improving existing Video-LLMs with the training data and shares helpful insights.

**Weaknesses:**

1.	The paper neglects to discuss many existing works that study the unanwerability of (V)QA models (see my attached references).

2.	It would be better to conduct more in-depth analysis to help understand what kinds of unanswerable QA are more challenging to resolve in videos: static objects/attributes/relations vs. dynamic actions.


[1] Whitehead S, Petryk S, Shakib V, et al. Reliable visual question answering: Abstain rather than answer incorrectly[C]//European Conference on Computer Vision. Cham: Springer Nature Switzerland, 2022: 148-166.

[2] Guo Y, Jiao F, Shen Z, et al. Unanswerable visual question answering[J]. arXiv preprint arXiv:2310.10942, 2023.

[3] Zhao Y, Zhang R, Xiao J, et al. Towards Analyzing and Mitigating Sycophancy in Large Vision-Language Models[J]. arXiv preprint arXiv:2408.11261, 2024.

[4] Brahman F, Kumar S, Balachandran V, et al. The art of saying no: Contextual noncompliance in language models[J]. arXiv preprint arXiv:2407.12043, 2024.

[5] Wen B, Yao J, Feng S, et al. The art of refusal: A survey of abstention in large language models[J]. arXiv preprint arXiv:2407.18418

**Questions:**

see weakness.

---

> ### Author Response · Authors · 2024-11-24
> **Author Response**
>
> ### [Response to Weaknesses & Questions]
> ### **Weakness 1: Missing references**
>
> Thank you for the valuable comment. Following the reviewer’s suggestion, we have added a new subsection, “Unanswerability in Question Answering”, in **[Section 2: Related Work]**. This subsection incorporates the references provided by the reviewer, discussing prior works on unanswerability in both LLM settings and image-based VLM settings.
>
> ---
>
> ### **Weakness 2: In-depth analysis of UVQA dataset by unanswerability type**
>
> Thank you for the comment. Following the reviewer’s suggestion, we have added an analysis section in **[Appendix 8: Performance Analysis by Unanswerability Type]**, which examines the performance of answerability across different categories of unanswerability.
>
> Specifically, we analyzed performance on three primary types of unanswerability: ***object***, ***relation***, and ***attribute***.
>
> For relation-related unanswerability, we further subdivided questions into two categories:
> 1. **Intra-object relationships** (involving a single object) and
> 2. **Inter-object relationships** (involving multiple objects).
>
> Both of these categories were further classified into **static** and **dynamic** relationships. (Examples of each category can be found in the paper).
>
>  Our findings indicate that object and relation-related unanswerable questions achieve comparable performance, whereas **attribute-related unanswerable questions pose a greater challenge.** *(Note that this difference in performance is not due to data imbalance in the unanswerable UVQA dataset, as we ensured the dataset was carefully balanced across these three categories during training, as noted in Section 5.1.)*
>
> On the other hand, for the subcategories of the relation-related unanswerable questions, we did not observe a consistent pattern where one type of relationship consistently outperformed or underperformed across different models.

---

> > ### Author Response · Authors · 2024-12-02
> > **Looking Forward to Your Reply**
> >
> > Dear Reviewer `yiLe`,
> >
> > As December 2nd is the final day for reviewers to provide comments, we would greatly appreciate it if the reviewer could review our rebuttals and provide any additional questions the reviewer might have.
> >
> > Thank you once again for the reviewer's time and thoughtful evaluation of our work.
> >
> > Best regards,
> >
> > Authors of Paper #10560

---

### Official Review · Reviewer_y4aQ · 2024-11-03

**Soundness:** 3
**Presentation:** 3
**Contribution:** 3
**Rating:** 6
**Confidence:** 3

**Summary:**

This paper studies the phenomenon that some video language models can't refuse to answer a question when the question is irrelevant to the video. The authors first showcase such failure cases in existing VLLMs, then provide a method and metric to identify such case, and finally propose a dataset to finetune existing VLLMs to improve this issue.

**Strengths:**

- This paper executed a valid pipeline of improving VLLM on a task: identifying the problem, curating a dataset to fix the problem, and finetuning the model to show improvements. The paper shows a good practice on the task of introducing refusing to answer for VLLMs.

- The alignment score defined in section 4 makes sense to me.

- The authors conducted experiments on a variety number of VLLMs in Table 1.

**Weaknesses:**

- My main feeling about reading this paper is I feel the problem of refusing to answer is a bit small and artificial. While showing the failure case in Figure 1, I feel it is also important to show if the problem can be migrated by explicit prompting, e.g., add to the prompt "Say 'can't answer' if you are not sure". Even though I believe the authors proposed method with finetuning on the curated dataset may still be better, I feel the problem is not as big as the author claimed.

- Adding to the above point, the authors only evaluated relatively small VLLMs in Table 1 (Video-LLaVA/ VideoChat2/ LLaMA-VID), while skipping the more recent SOTA VLLMs like Qwen2-VL, LLaVA-OneVision, and GPT-4o or Gemini 1.5. I am expecting these more well instruction-tuned models may have less answerability issues.

- If I understand correctly, the authors propose to solve the problem by finetuning the VLLM on a dataset. A clear shortcoming is that the models might drop performance on other tasks, which is not desirable. Please provide some discussion on this.

**Questions:**

- Overall, I feel this paper propose a valid solution to a small problem. The solution might have issues on overfitting the model on the particular task, and the problem may not exist on more modern VLLMs. My current rating is a week reject. I am happy to raise my rating if the authors address my concerns in the rebuttal.

---

> ### Author Response · Authors · 2024-11-24
> **Author Response**
>
> ### [Response to Weaknesses & Questions]
>
> ### **Weakness 1-(a): The problem of refusing to answer feels a bit small and artificial.**
> We appreciate the reviewer’s feedback. While the problem of refusal behavior may initially seem small or artificial, we believe it has significant implications for the application of Video-LLMs in real-world scenarios. In particular, Video-LLMs are increasingly being considered for use in assistive technologies, such as answering questions about streaming video content for visually impaired or blind individuals. In these contexts, ensuring that the model appropriately refuses to answer when uncertain or when the input exceeds its capabilities is critical for maintaining user trust and preventing misinformation.
>
> ---
>
> ### **Weakness 1-(b) & 2: Does the problem persist on SOTA Video-LLMs with bigger model sizes, and can the problem be mitigated by using an unanswerability prompt**
>
> Thank you for the insightful comments and suggestions. To address the reviewer's concerns, we have conducted a series of additional experiments, detailed below.
>
> **[Does Model Size Affect Answerability?]**
>
> In Figure 1-(b), we have added a comparison between a 7B model and a 72B model across different variants of open-sourced Video-LLMs (LLaVA-OneVision [1] and LLaVA-Video-qwen2 [2]). While scaling up model size improves performance on traditional answerable QA benchmarks, we observe that the inability of these models to handle unanswerable questions persists regardless of model size. This indicates that scaling alone does not address the issue of unanswerability.
>
> ```
> [1] Li, et al. LLaVA-OneVision: Easy Visual Task Transfer, arxiv 2024
> [2] Zhang, et al. Video Instruction Tuning With Synthetic Data, arxiv 2024
> ```
>
> **[Can Prompting Improve Answerability?]**
>
> To explore the reviewer’s suggestion, we conducted experiments on whether explicitly adding the prompt, "If the question cannot be answered using the video content, state that it is unanswerable and provide a reason.” can improve the handling of unanswerable questions by open-sourced VLMs. These results are included in ***[Appendix A.9: Can Prompting Detect Unanswerability?]***. We find that explicit prompting does not significantly improve answerability performance for these models, regardless of their size. This underscores the importance and relevance of the problem we address in this work.
>
> **[How About Closed-Sourced VideoLLMs Like GPT-4o?]**
>
> Our main experiments focused on open-sourced Video-LLMs, as described in the paper. To provide a broader perspective, we extended our analysis to include a closed-source Video-LLM (GPT-4o). Interestingly, GPT-4o demonstrated some ability to handle unanswerable questions, achieving an accuracy of 0.50 on the UVQA dataset, compared to 0.57 by the DPO model of VLM-RLAIF (7B) trained using our framework. This suggests that GPT-4o likely underwent internal alignment processes for answerability, as open-sourced models without such alignment consistently fail in this area. Alignment is particularly crucial for closed-source models deployed as services, as effective handling of unanswerable questions is essential for trustworthiness and safety.
>
> While closed-source models may address this issue internally, their methods remain undisclosed. Our work emphasizes the limitations of open-sourced Video-LLMs in handling unanswerable questions and provides an open-source dataset and framework to tackle this challenge.
>
> ---
>
> ### **Weakness 3: Trade-offs between answerability performance and standard VideoQA performance.**
>
> As is often the case in multi-objective optimization, there is a trade-off in our setup as well. This trade-off is an inherent limitation, which we have explicitly acknowledged and addressed in our work. The metrics we propose, such as  $S\_{\text{ex-ref}}$ (Eq. 5), are designed to evaluate models while considering such trade-offs. For example,  $S\_{\text{ex-ref}}$  quantifies the extent to which models, after alignment, decline to answer answerable questions that it was previously capable of answering correctly. A lower $ S_{\text{ex-ref}}$  score is better, and unaligned models naturally achieve a score of 0. However, after alignment through SFT or DPO, models exhibit  $S\_{\text{ex-ref}}>0$, indicating some trade-off in performance on previously answerable questions.
>
> To further address this, we added a section in **[Appendix A.10: Pareto Front Analysis]** to comprehensively illustrate the trade-off. Specifically, we observe that alignment for answerability improves accuracy on the unanswerable UVQA dataset (**0.0 $\rightarrow$ 0.57**) but slightly decreases accuracy on the answerable QA dataset (**0.49 $\rightarrow$ 0.47**). Notably, the Pareto front analysis demonstrates that the aligned models (SFT, DPO) achieve a better balance, positioning them at an optimal point where performance on unanswerable questions improves without excessively compromising performance on answerable questions.

---

> > ### Comment · Reviewer_y4aQ · 2024-11-25
> > **Main concerns resolved**
> >
> > Thank the authors for providing the detailed rebuttal and especially providing the experimental results.
> >
> > I had 3 main concerns before the rebuttal. 1. Significance of the task; 2. Does the problem have simpler solutions (better prompt/ more advanced models); 3. VLM performance drop after the proposed finetuning.
> >
> > Concern 2 is fully addressed by the authors additional experiments on LLaVA-OneVision and GPT4o, and the prompt experiments. Concern 3 is also well addressed by evaluating on answerable QA dataset (more datasets will make it even better though). While I still have concerns on the significance of the task, I appreciate the authors effort in the rebuttal. I believe the authors did a great job in validating their contributions under the current setting. I'm happily raising my score to a weak accept.

---

### Official Review · Reviewer_GEbz · 2024-11-04

**Soundness:** 3
**Presentation:** 4
**Contribution:** 3
**Rating:** 6
**Confidence:** 4

**Summary:**

This paper introduces the concept of alignment for answerability in the context of video LLMs. Current models and data do not consider out-of-scope questions that a user might ask, and hence they don't recognize them as unanswerable. A definition and a metric to assess this phenomenon are presented. Furthermore, a new dataset is proposed containing unanswerable questions, and fine-tuning models on this dataset shows improved alignment towards recognizing unanswerable questions.

**Strengths:**

The paper is well-written and clearly introduces the concepts, definitions and metrics. The contribution is original as it formally recognizes a new problem, i.e. multimodal LLMs answering clearly unanswerable questions, and proposes solutions to it.
- The proposed problem framing and metric make intuitive sense. The definition takes into account the reasoning of why a question is unanswerable. The metric takes into account excessive refusal to answer which is a big problem with LLMs.
- A new dataset for alignment for answerability is presented. This is a valuable addition to the community as it creates out-of-scope QA pairs which we rarely see.
- The dataset creation process is simplistic, modifying a constrained set of objects and their relations, or changing object attributes. Furthermore, the evaluation dataset is verified manually. So, the dataset is likely dependable.
- Experimental results after fine-tuning on this dataset show clear improvements in alignment (Table 1), and the ablation study in Figure 5 outlines the drawback of an alternative.

**Weaknesses:**

The main weakness is that the proposed QAs in the dataset seem to be mostly geared towards detection capabilities without requiring much reasoning. This in turn makes the corresponding dataset construction, and improving model's capabilities on this axis rather easy. Some qualitative examples presented are "What breed is the cat in the video?", "What color laptop the presenter is holding?", "How many times does a person in gray shirt appear in the video?", etc. Recently, many reasoning based video-language benchmarks are proposed [1, 2] while this paper is more similar to older benchmarks [3].

Other notable weaknesses:
- The metric is rather complex. If we just want to detect unanswerable questions, and balance precision and recall, why not use F1-Score?
- The metric is defined between aligned model and unaligned model. However, if one wants to test the ability of a given model regarding unanswerable questions, the proposed metric cannot be used.
- Using LLM-in-the-loop to create the dataset may create some bias, hallucinations and incorrect QA pairs.
- Human performance on the evaluation set is not presented.

[1] Perception Test: A Diagnostic Benchmark for Multimodal Video Models \
[2] MVBench: A Comprehensive Multi-modal Video Understanding Benchmark \
[3] ActivityNet: A large-scale video benchmark for human activity understanding

**Questions:**

Can the authors provide more qualitative examples of unanswerable questions from the proposed dataset? Similarly, can we also get more qualitative examples of existing & aligned model's reactions to such questions?

---

> ### Author Response · Authors · 2024-11-24
> **Author Response 1/N**
>
> ### [Response to Weaknesses & Questions]
> ### **Weakness 1: The proposed QAs in the dataset seem to be mostly geared toward detection capabilities without requiring much reasoning.**
>
> We appreciate the reviewer’s observation and would like to clarify that our dataset does include more complex question forms than the examples originally shown in the paper. In ***Appendix A.4.3***, we have added more qualitative examples of unanswerable questions from our proposed UVQA dataset.
>
> For instance, the examples include questions such as:
> "*Why is the man with the blue-green jacket pointing at something in the video?*", "*In which sequence does the pug eat from a blackish dog dish in the video?*", which are question forms that require contextual understanding and temporal understanding, respectively.
>
> **However**, as the reviewer noted, determining unanswerability for these questions still primarily relies on the detection capabilities of the Video-LLM. ***We would like to note that because we are dealing with questions that go beyond the information boundaries of the video, it is natural that we rely on the detection capabilities of the Video-LLMs to discern these questions.***
>
> That said, we believe it is important to emphasize that, as shown in Figure 1-(a) and (b), open-sourced Video-LLMs, regardless of parameter size (up to 72B), consistently fail to address even these detection-based unanswerable questions. This highlights a significant pitfall in current models’ capabilities, even for what might seem like simpler unanswerable questions.
>
> We also agree with the reviewer that extending this work to include unanswerability requiring more advanced reasoning would be a valuable future direction. Our work aims to provide a foundational step toward this goal, setting the groundwork for future research to tackle increasingly complex unanswerable scenarios.
>
> ---
> ### **Weakness 2: The metric is rather complex. Why not use F1-Score?**
>
> **[Why we didn't use F1-score]**
>
>  We chose not to use the F1-score because our evaluation framework extends beyond a binary classification of questions as simply answerable or unanswerable. Specifically, if the model predicts a question as unanswerable, we only consider the prediction correct if the reasoning for unanswerability is also correct. This distinction is reflected in Figure 2, where we differentiate between "unanswerability correct" ($\text{unanswerable}\_{\text{c}}$) and "unanswerability wrong" ($\text{unanswerable}\_{\text{w}}$). Similarly, when the model predicts a question as answerable and provides an answer without any "unanswerable" indicators, correctness is determined by how well the generated response aligns with the ground truth answer.
>
> However, we agree that a simplified evaluation of binary correctness for answerable vs. unanswerable predictions can offer additional insights. **Therefore, we have included this metric, using F1-score, in the revised manuscript (Table 1)**.
>
> &nbsp;
>
> **[Why accuracy alone is insufficient]**
>
> Moreover, we would like to clarify why we proposed a seemingly complex metrics (Eq. 5-7) rather than solely evaluating on the accuracy value (Eq. 4). Accuracy, as a standalone metric, fails to capture essential details introduced by the alignment process. For instance, consider an answerable question $Q_A$:
> 1. The unaligned model correctly answers $Q_A$.
> 2. After alignment for answerability, the aligned model either incorrectly answers $Q_A$ or predicts it as unanswerable.
>
> A naive accuracy measurement cannot differentiate whether the aligned model's error arose from an incorrect answer or an incorrect "unanswerable" prediction. This limitation makes it difficult to analyze how alignment (e.g., the alignment algorithm $f$ in Eq. (3) or the dataset used for alignment) impacts the model's performance.
>
> This motivated us to devise a metric that provides a more detailed and explainable evaluation of the alignment process. **Our proposed metric captures various aspects, including the case described above, offering deeper insights into the alignment's effects on the model.**

---

> ### Author Response · Authors · 2024-11-24
> **Author Response 2/N**
>
> ### **Weakness 3: The metric is defined between aligned model and unaligned model. However, if one wants to test the ability of a given model regarding unanswerable questions, the proposed metric cannot be used.**
>
> To evaluate the performance of a given model, we can use the straightforward accuracy metric $S\_\text{acc}$ defined in Eq. (4). This metric calculates the overall accuracy by considering how well a given model responds to both the answerable and unanswerable questions. However, it can also be decomposed to focus on these two cases separately:
>
> $$
> S\_{\text{acc, ans}} = \frac{N_1 + N_4 + N_7}{N_{1-9}}
> $$
>
> $$
> S\_{\text{acc, unans}} = \frac{N_{12} + N_{15} + N_{18}}{N_{10-18}}
> $$
>
> Here, $S\_{\text{acc, ans}}$ measures the accuracy of a given model on answerable questions, while $S\_{\text{acc, unans}}$ measures the accuracy of a given model on unanswerable questions. Along with our proposed metric defined between unaligned and aligned models, we also use these kinds of **absolute performances**, which evaluate the performance of a single given model, in our main result (Table 1).
>
> ---
>
> ### **Weakness 4: Using LLM-in-the-loop to create the dataset may create incorrect QA pairs.**
>
> While using an LLM-in-the-loop to create the dataset may introduce some noise, we have implemented filtering processes to mitigate issues such as incorrect QA pairs, as it had been detailed in **[Appendix A.4.2: Dataset Filtering]**.
>
> For the **training dataset**, we perform an **automatic filtering** step to remove instances where the altered description is too semantically similar to the original description. This step minimizes the risk of incorrect labeling, where the generated unanswerable question remains answerable.
>
> For the **evaluation dataset**, we apply a **human filtering** process to ensure high quality. Specifically, we manually review all samples and remove any questions that remain answerable after the alteration process. This ensures that the evaluation dataset reflects the intended characteristics of unanswerable questions.
>
> These steps collectively reduce noise and improve the quality of both the training and evaluation datasets.
>
> ---
>
> ### **Weakness 5: Human performance on the evaluation set is not presented.**
>
> We have used the **$\text{LLM}\_\text{score}$** in Table 1 as a proxy for human evaluation, following a common practice in recent research where LLMs are used to evaluate generated content. This approach has been shown to correlate strongly with human preferences, as demonstrated in [LLM-as-a-Judge](https://arxiv.org/abs/2306.05685) [1].
>
> That said, we agree that explicitly including a human evaluation is a valuable addition to the results. In response to your suggestion, we are currently in the process of performing a human evaluation. **We will update on this once they are finished.**
>
> > [1] Zheng et al., Judging LLM-as-a-Judge with MT-Bench and Chatbot Arena,NeurIPS 2023 Datasets and Benchmarks Track
>
> ---
>
> ### **Question 1: More qualitative examples of UVQA dataset and prediction results of existing \& and aligned models**
>
> Thank you for the suggestion. In response to your feedback, we have added more qualitative examples of unanswerable questions from the proposed UVQA dataset in **[Appendix A.4.3: Examples of the Generated UVQA Dataset]**. Additionally, we have included more qualitative examples of the
> predicted responses from both the existing and aligned models in **[Appendix A.7: Additional Examples of Model Predictions]**.

---

> ### Author Response · Authors · 2024-11-26
> **Author Response (Follow Up on Human Evaluation)**
>
> As detailed in our response to Weakness 5, we have completed the human evaluation and included the results in ***[Appendix A.11: Human Evaluation of Model Prediction]***. The results show a high correlation between the human evaluation and the **$\text{LLM}\_\text{score}$**, further validating the use of an LLM evaluation as a proxy for human evaluation. Additionally, the human evaluation confirms that the aligned model, using our framework, outperforms the unaligned model.
>
> We sincerely thank the reviewer once again for the insightful feedback.

---

> > ### Comment · Reviewer_GEbz · 2024-11-29
> >
> > Thanks for the clarifications, I have no further questions. Overall, I think this is a valuable addition to the community.

---

### Author Response · Authors · 2024-11-24
**General Response to Reviewers**

We would like to thank all the reviewers for the detailed review and feedback. We have addressed each noted question separately. We have uploaded a revised version of the manuscript with the following additions (**revised parts are indicated in blue in the manuscript**):

* **Figure 1-(b)** has been added, which shows that scaling the model  (7B $\rightarrow$ 72B) does not improve the answerability performance.

* *Unanswerability of Question Answering* section has been added to **Section 2: Related Work**.

* In **Table 1**, we have added binary correctness for answerable vs. unanswerable predictions using the **F1-score**.

* In **Appendix A.4.3**, we added more qualitative examples of the generated UVQA dataset.

* **Appendix A.7** has been added to show more qualitative examples of model predictions.

* **Appendix A.8** has been added to analyze the performance difference with respect to the unanswerability type.

* **Appendix A.9** has been added to show if prompt addition alone (such as adding the prompt `If the question cannot be answered using the video content, state that it is unanswerable and provide a reason.') can achieve alignment for answerability.

* **Appendix A.10** has been added to visualize the trade-off between the answerable QA accuracy and the unanswerable QA accuracy using the Pareto front visualization.

---

> ### Author Response · Authors · 2024-11-27
> **Additional Revisions to the Manuscript**
>
> * **Appendix A.11** has been added to show the human evaluation of model predictions.
> * **Appendix A.12** has been added to show the performance on the out-of-distribution evaluation set, which uses videos from a different source and includes fully human-annotated unanswerable QA pairs.
>
> *We sincerely thank the reviewer for their valuable time, thoughtful comments, and constructive feedback, which have greatly helped improve our work.*

---

### Meta-Review · Area_Chair_eFMx · 2024-12-21

**Metareview:**

The authors address a timely and relevant task to the community: Enabling VLMs to refuse an answer when there is not sufficient information in the video. Reviewers had some initial concerns about the paper (ie degradations on existing tasks, human evaluations, how much is this really a problem in practice), but the authors addressed these comprehensievely in the rebuttal. As a result, all the reviewers were in favour of accepting the paper. Thank you for updating and improving the paper during the rebuttal phase.

**Additional Comments On Reviewer Discussion:**

Please see above. Reviewers had some initial concerns about the paper (ie degradations on existing tasks, human evaluations, how much is this really a problem in practice), but the authors addressed these comprehensievely in the rebuttal.

---

### Decision · Program_Chairs · 2025-01-22

Accept (Poster)